# Factors Influencing Venous Remodeling in the Development of Varicose Veins of the Lower Limbs

**DOI:** 10.3390/ijms25031560

**Published:** 2024-01-26

**Authors:** Lukasz Gwozdzinski, Anna Pieniazek, Krzysztof Gwozdzinski

**Affiliations:** 1Department of Pharmacology and Toxicology, Medical University of Lodz, 90-752 Lodz, Poland; lukasz.gwozdzinski@umed.lodz.pl; 2Department of Oncobiology and Epigenetics, Faculty of Biology and Environmental Protection, University of Lodz, 90-236 Lodz, Poland; anna.pieniazek@biol.uni.lodz.pl

**Keywords:** chronic venous insufficiency, varicose vein, pathophysiology, inflammation, endothelial dysfunction, oxidative damage, flavonoids, saponins, pycnogenol, ginkgo biloba

## Abstract

One of the early symptoms of chronic venous disease (CVD) is varicose veins (VV) of the lower limbs. There are many etiological environmental factors influencing the development of chronic venous insufficiency (CVI), although genetic factors and family history of the disease play a key role. All these factors induce changes in the hemodynamic in the venous system of the lower limbs leading to blood stasis, hypoxia, inflammation, oxidative stress, proteolytic activity of matrix metalloproteinases (MMPs), changes in microcirculation and, consequently, the remodeling of the venous wall. The aim of this review is to present current knowledge on CVD, including the pathophysiology and mechanisms related to vein wall remodeling. Particular emphasis has been placed on describing the role of inflammation and oxidative stress and the involvement of extracellular hemoglobin as pathogenetic factors of VV. Additionally, active substances used in the treatment of VV were discussed.

## 1. Introduction

Chronic venous disease (CVD) is associated with a disease of the veins of the lower extremities. The veins develop into varicose veins (VV), which are characterized by the presence of dilated, bulging, twisting veins beneath the surface of the skin. In addition, there are spider veins that may surround the varicose veins. These are smaller red or purple blood vessels near the surface of the skin [1]. It is believed that the formation of varicose veins is caused by valve dysfunction and venous reflux, which consequently leads to venous stasis and hypoxia. Left untreated, VVs can lead to the formation of venous leg ulcers (VLU) [2]. However, in some cases, they can lead to serious health problems such as thromboembolism. A common cause of varicose veins is chronic inflammation, which may have a genetic basis and can cause damage to the valves in the veins of the lower limbs [3]. Blood stasis in varicose veins and hypoxia can lead to cells releasing inflammatory mediators and growth factors through endothelial cells (ECs). Inflammatory mediators attract and activate neutrophils, leading to their infiltration of the venous wall, and initiate damage to extracellular matrix components. In contrast, growth factors cause migration, proliferation, and dedifferentiation of smooth muscle cells, which stimulates the formation of neointima [4]. As a consequence, there is stagnation of blood flow, hypoxia, and further damage to the vein causing the formation of varicose veins. It has been shown that varicose veins are characterized by a greater infiltration of inflammatory cells compared to the normal vein. Macrophages/monocytes and mast cells participate in the damage and remodeling of the vein [5,6]. Numerous studies have shown that growth factors secreted by macrophages, pro-inflammatory cytokines, matrix metalloproteinases (MMPs), and adhesion molecules are involved in the etiology of varicose veins [7]. Of all white blood cells, macrophages/monocytes are most numerous in damaged veins of the lower limbs. Compared to healthy veins, these cells are located near valves and vein walls [7,8]. They were found in adventitia and vasa vasorum but not in the muscular layer [9]. These cells can release proteolytic enzymes and reactive oxygen species (ROS), participating in damage to valves and venous walls [8]. It has been shown that the neutrophils of patients with a chronic venous disease produce more ROS compared to healthy individuals [10,11]. Pro-inflammatory factors, including ROS and matrix metalloproteinases, are released by the vascular endothelium, leading to the accumulation of neutrophils and platelets at the site of inflammation. Inflamed endothelial cells express cell adhesion molecules such as E-selectin, P-selectin, and von Willebrand factor (vWf). Neutrophils and platelets form bonds with these molecules, which initiates an inflammatory cascade and in consequence leads to thrombosis [12].

Factors favoring the development of VV include familial and genetic predisposition associated with CVD, as well as other factors such as behavioral and dietary factors, including obesity and a sedentary lifestyle. Moreover, the etiology and pathophysiology of chronic venous disease include genetic, proteomic, hormonal, and cellular mechanisms that influence the changes induced in the structure and functioning of venous vessels. The expression of several genes related to angiogenesis, vascular hyperplasia, and venous regulation influences the incidence of CVI [13]. In addition, inflammation, changes in mRNA expression, protein levels, and the proteolytic activity of matrix metalloproteinases (MMPs) have been found in VV and VLU [2].

Depending on the severity of the disease, varicose veins can be treated with compression stockings, oral medications, laser therapy, sclerotherapy, or surgery [14]. There are various types of surgical treatment for sealing pathological venous vessels. Some of these methods, such as endovascular laser or radiofrequency ablation use thermal ablation, others mechanical ablation, using a special catheter and foam or chemical ablation using special foam in a similar way to sclerotherapy. Pharmacological treatment, used at all stages of CVD and together with other methods, involves the use of venotropic drugs of plant origin as well as synthetic drugs. However, if left untreated, varicose veins can lead to serious consequences, such as venous leg ulcer development and thrombosis, including deep vein thrombosis (DVT). Altered venous hemodynamic triggers proteolytic remodeling of the venous wall and inflammatory processes as well as degradation of the protective endothelial glycocalyx, resulting in a wide spectrum of clinical symptoms ranging from varicose veins to venous ulcers, which have been termed chronic venous disorders (CVeD).

This review presents the various anomalies in the structure and function of the lower limb veins observed in CVeD. The etiology and pathogenesis of CVeD are discussed, including genetic and environmental factors that may contribute to an increased risk of CVeD. We then describe the role of the endothelium in VV pathophysiology, mechanisms related to VV hypoxia, and factors influencing MMP modulation and activity. The role of blood cells in this disease, the role of oxidative stress (OS) and inflammation in vascular wall damage, and the role of OS in damage to red blood cells and the endothelium are also presented.

## 2. Chronic Venous Insufficiency

### 2.1. Factors Influencing the Occurrence and Development of CVeD

Varicose veins are a very common disease, affecting about 1/3 of the adult population. They occur more often in women than men. The factors contributing to the occurrence of varicose veins include the general health and the age of patients (over the age of 50), which is related to the aging of the vein walls (the veins lose elasticity and stiffen) and valves, which no longer function as efficiently as at a younger age. In addition, varicose veins can arise as a result of a sedentary lifestyle, obesity, and smoking. No less important are familial predispositions, and the occurrence of the disease in the family. In the case of women, female hormones can lead to stretching of the walls of the veins. In addition, taking birth control pills can aggravate this process. Menopause is also a factor contributing to the disease. The most common symptoms of chronic venous disease of the lower extremities are pain, heaviness, leg fatigue, cramps, itching, burning sensation, swelling, and restless legs syndrome. In addition, visible changes in the lower extremities are skin changes, induration of subcutaneous fat (lipodermatosclerosis), telangiectasias, reticular veins and varicose veins, edema, pigmentation, and ultimately ulceration [15].

### 2.2. Changes in the Vein Wall Structure

Chronic venous insufficiency is caused by abnormalities in the wall structure and dysfunction of the venous valves, as well as disorders resulting from previous deep vein thrombosis [16]. Disturbances in the structure of the vein wall are related to the abnormal extensibility of the connective tissue in the vein wall. It has been shown that the veins of patients with varicose veins differ, among other things, in their elastic properties compared to the veins of healthy people [17]. Primary varicose veins are a consequence of venous dilatation and valvular failure without previous deep vein thrombosis. Secondary varicose veins are caused by DVT or superficial thrombophlebitis [18]. In the vessels of the lower limbs, the calf muscle pump, called the peripheral heart, plays an important role influencing proper blood flow and preventing venous stasis.

With properly functioning valves, contraction of the calf muscle compresses the vein and pumps blood upwards [19]. During walking, the calf muscle pump causes a decrease in the capacity of the venous system, which leads to a decrease in the pressure in the veins. In turn, the relaxation of the muscle pump causes the veins to refill with blood. Dysfunction of the valves of the venous system leads to a reverse flow of blood, i.e., venous reflux, and the development of chronic venous insufficiency. This process concerns superficial and deep veins, perforating veins, and venous tributaries [20]. As the insufficiency increases, various changes are observed at the level of the venous walls. Disturbances in the types of collagen in the venous wall affect not only the fibroblasts in the vein walls but also the skin fibroblasts of patients with varicose veins [21]. Moreover, genetic influence on the remodeling of vessel walls has been demonstrated, which is associated with imbalance in tissue MMPs and their inhibitors and a decrease in the expression of desmuslin in smooth muscle cells, as well as thrombomodulin mutations 1208/1209TT [2,21]. It has been shown that the ratio of collagen I to collagen III is disturbed in CVD patients, i.e., there is an increase in the collagen I/collagen III ratio and MMP-2, and the TIMP-2 expression levels are reduced [2,21].

Changes in shear stress leading to activation, leukocyte adhesion, and migration across the endothelium contribute to inflammation and subsequent remodeling of the venous wall and valves [22]. Pathological shear stress with multidirectional orientation can cause the expression of atherogenic and thrombogenic genes, and accelerate endothelial cell proliferation and turnover. In addition, it has pro-inflammatory, pro-coagulant, pro-oxidative, and pro-apoptotic effects, which consequently lead to endothelial dysfunction [23,24].

### 2.3. The Role of Inflammation in CVD

The main mechanism associated with the pathophysiology of chronic venous insufficiency is an increase in venous pressure, which is a consequence of damaged venous valves, shear stresses, and reflux [25]. These factors cause further damage to the valves, increasing pressure and dilating the vein. Changes in the vein are transferred to microcirculation, disturbing the function of endothelial cells and the vascular microenvironment, which in turn leads to venous microangiopathy, characterized by dilatation and tortuosity of the capillary beds [25]. Changes in vessel hemodynamics activate various biological processes, such as inflammation, proteolytic enzymes release in the vascular microenvironment, as well as leukocyte adhesion, degranulation, and the release of granules from neutrophils, mastocytes, endothelial cells, and platelets [26].

In patients with varicose veins, the presence of many inflammatory molecules, cytokines, chemokines, vasoactive factors, selectins, and prothrombotic precursors has been demonstrated, including adhesion molecules: ICAM-1, VCAM, chemoattractant protein MCP-1, L-selectin, E-selectin, cytokines: IL-1β, IL-4, IL-6, IL-8, IL-12, IL-13, p40, G-CSF, GM-CSF, IFN-γ, TNF-α, MIP-1a, and matrix metalloproteinases MMP-1, -2, -3, -8, -9, -12, and -13, and macrophage inflammation protein 1b [27]. Platelets reach sites of inflammation and can lead to coagulation as well as an immune response promoting leukocyte–endothelial interaction. Thus, platelets are involved in the pathology of inflammation [28].

### 2.4. Classification System for Chronic Venous Disorders

The CEAP (clinical–etiology–anatomy–pathophysiology) classification has been adopted throughout the country to enable the assessment of the CVD status and the use of appropriate treatment methods [29]. The pathophysiology of CVD is characterized by different clinical stages, starting with the clinical class C0s, which includes patients without visible or palpable symptoms of a venous disease. Between 13% and 23% of the general population belong to the C0s class. The main cause of CVeD is remodeling of the vein wall and damage to the valves, which is a consequence of blood stasis, hypoxia, and interaction of the endothelium with white blood cells. The formation of varicose veins can cause changes leading to prothrombotic syndrome followed by deep vein thrombosis. In turn, skin changes and ulcerations are caused by venous hypertension, which is transferred to the microcirculation [30]. In cutaneous microcirculation, changes are progressive in groups C1 to C6. It was also found that the diameter of the capillaries increased and their morphology deteriorated in groups from C2 to C5. In turn, an increase in the diameter of the dermal papilla, as well as in the diameter of the volume of capillaries, appears in groups C3 to C5. In contrast, the functional capillary density (FCD) decreases from groups C4 to C5 [30]. The presence of varicose veins may lead to a prothrombotic state and then to deep vein thrombosis.

## 3. Vascular Endothelium

In the pathogenesis of vascular lesions, an important role is performed by the vascular endothelium, which directly interacts with blood cells. One of the functions of the endothelium is to concentrate various biochemical and biomechanical signals to maintain a barrier function providing selective permeability, vascular tone, proper blood viscosity by regulating laminar flow, and the formation of new vessels. The endothelium covering the inner surface of blood vessels is in constant contact with the morphotic components of blood, but also with its metabolites and immune cells, being responsible for the regulation of homeostasis [31]. In the presence of inflammation, the vascular endothelium is exposed to many factors, for example, ROS, proteases, and others, which may lead to its dysfunction or damage.

### 3.1. Vascular Endothelium in Inflammation

During inflammation, ROS are overproduced. On the one hand, excessive ROS production inhibits the release of nitric oxide (NO·), which affects vascular stiffness and contractility, leading to endothelial dysfunction [12]. However, the overproduction of NO leads to oxidative stress and cell apoptosis [32]. Many studies have shown a significant contribution of nitric oxide synthases (mainly iNOS and eNOS) and nitric oxide in ROS activity in vascular diseases. In addition, endothelial cells can regulate gene expression. For example, class A genes include E-selectin, thrombomodulin, endothelial protein C receptor, endothelial isoforms of nitric oxide synthase (eNOS), endocan, and vWf. In turn, the vascular endothelial growth factor (VEGF) released by platelets and other cells regulates endothelial cell (EC) proliferation and migration. Endothelium is also involved in the processes of aging, autophagy, and cell death, and releases signaling molecules such as nitric oxide, prostanoids, and others [22]. It participates in inflammation initiated by infection or tissue damage. Inflammatory factors interacting with ECs lead to impaired vascular tone, increased permeability, increased procoagulant dynamics, and impaired vascular formation mechanisms. A dysregulated mechanism related to the inflammatory response is associated with the development of vascular diseases.

### 3.2. The Role of Adhesion Proteins and Cytokines in Inflammation

Non-activated endothelial cells express intercellular adhesion molecules (ICAM). On the other hand, E-selectin and vascular cell adhesion molecules (VCAM) are found only in stimulated ECs. In inflammatory conditions, cytokines TNF-α, IL-1β, and IFN-γ, acting on endothelial cells, can initiate the synthesis and expression of E-selectin and VCAM, and also lead to an increase in the level of ICAM [33]. In the adhesion of leukocytes to the endothelium under flow conditions, interactions between selectins and their respective ligands perform an important role [34]. In inflammation and infection or vascular damage, the surface of the endothelium becomes susceptible to the adhesion of leukocytes, which is an innate immune response. This process is regulated and consists of a multi-step cascade that includes successive stages of interaction between leukocytes and the endothelium. Selectins are involved in the initial phase, and after leukocyte activation by chemokines present on the endothelial surface, leukocyte integrins (CD11/CD18) are activated, leading to leukocyte adhesion to the endothelium. In addition, platelets circulating in the bloodstream may participate in the coagulation process as well as in the immune response [28]. The basic integrin ligands that participate in leukocyte adhesion include intercellular adhesion molecules 1–5, vascular cell adhesion molecule-1, and junctional adhesion molecules (JAMs) expressed on endothelial and other cells [35,36]. In addition, the important adhesion receptors involved in leukocyte recruitment include platelet endothelial cell adhesion molecule-1 (PECAM-1) and endothelial cell adhesion molecule (ESAM) [37,38].

Proinflammatory cytokines and adhesion molecules perform important roles in the development of venous thrombosis. In inflammation, neutrophils aggregate and adhere to ECs [39,40]. Initially, P, E, and L selectins participate in the process of adhesion to the endothelium [41]. P-selectin, in particular, performs an important role in diseases associated with damage and thrombosis of blood vessels, selectin initiates the accumulation of leukocytes and adhesion to the endothelium and then in the accumulation of platelets [42]. However, stronger binding to the endothelium occurs through the CD11/CD18 complex and the ICAM-1 and ICAM-2 [43,44]. Adhesion can also be induced by cytokines (IL-8 and IFN-γ), platelet activation factor (PAF), as well as the active complement complex and arachidonic acid metabolites [42,45]. Activated neutrophils produce reactive oxygen species and proteases [46]. On the other hand, TNF-α causes increased activation of neutrophils, their adhesion, degranulation, and the production of ROS, which occurs in the presence of p55 and p7 proteins [47,48].

### 3.3. Cellular Response in Inflammatory Conditions

There is chronic inflammation in varicose veins as indicated by elevated levels of inflammatory and prothrombotic markers [49,50]. The consequences of chronic inflammation after vascular damage are pathological interactions of the activated endothelium, neutrophils, and platelets, as well as fibrosis and thrombosis [12]. In the case of varicose veins, the developing disease causes damage to the vein, which can lead to the formation of blood clots. Inflammation accompanies the pathophysiology of deep vein thrombosis, pulmonary embolism, and peripheral arterial disease [51]. Patients with varicose veins have a significantly increased risk of DVT [52].

Cells respond to the influence of the environment, which may lead to changes in morphology, mobility, or proliferation, affecting their state of differentiation [53]. The structure and condition of the endothelial cytoskeleton performs an important role in leukocyte recruitment and fibrosis. In endothelial cells stimulated by TNF-α, there is a gradient of cortical stiffness of rolling neutrophils, which causes them to be directed to sites of transmigration [54]. Pro-fibrotic promoting factors, e.g., autotaxin (ATX), responsible for the production of lysophosphatidic acid, lead to the rearrangement of endothelial actin and cell contractility, which causes vascular leakage, leading to the migration of fibroblasts in the underlying tissue [55]. Moreover, changes in the structure of the cytoskeleton contribute to changes in the distribution and function of the glycocalyx [56], which in turn further deepens endothelial dysfunction and leads to vascular damage.

Using two fluorescent probes located at different depths of the lipid monolayer of the membrane, we demonstrated lower membrane fluidity of human varicose vein endothelial cells (HVVEC) in comparison to the human umbilical vein endothelial cells (HUVEC) in the subsurface area of the membrane. Greater differences in membrane fluidity were found for the probe located in the deeper region of the monolayer. These results demonstrated a higher stiffness of HVVEC plasma membranes compared to HUVEC cells. Differences in the fluidity of normal and pathological membranes may be the result of lipid–lipid and lipid–protein interactions or may be caused by oxidative stress occurring in the pathological vein [57]. Changes in membrane fluidity are closely related to the passive and active transport of substances across membranes into and out of the cells [58,59]. Previous studies have shown that the fluidity of the plasma membrane of the cells performs a significant role in the mechanism of cell adhesion [60,61]. Membrane fluidity is also an important parameter determining communication between cells [62,63]. The mechanism of cell adhesion is improved by building nanoclusters, ordered lipid rafts heterogeneously distributed in the membrane, which are associated with adhesive complexes [64]. A more fluid environment allows the rapid dispersal of lipid rafts, which leads to a reduction in adhesions [60,61]. The reduced fluidity of HVVEC may result in increased adhesion properties and, therefore, a greater likelihood of venous thrombosis. Interestingly, drugs used in varicose vein therapy such as diosmin and aescin, as well as bromelain, lead to a slight increase in fluidity in the near-surface region of the lipid bilayer of the membrane and in consequence to their lower adhesive properties [57].

## 4. Hypoxia in Varicose Veins and Its Consequences

McEwan and McArdle showed significantly lower oxygen levels in varicose blood than in non-varicose blood [65]. The oxygenation of the venous wall occurs through the diffusion of oxygen from the luminal blood and vasa vasorum. It turned out that the oxygen concentration was lower in the venous wall of the VV than in the non-varicose veins. [66]. Hypoxia induces inflammation and oxidative stress as well as activation of leukocytes and endothelial cells [67]. These cells release many factors that affect the regulation of venous wall remodeling. It is believed that hypoxia is one of the factors initiating the formation of varicose veins. In addition, leukocyte adhesion, caused by low oxygen concentration, leads to increased production of reactive oxygen species. It has been suggested that leukocyte–endothelial interactions are caused by a change in the ROS-nitric oxide balance in hypoxia [68].

### 4.1. The Expression of Hypoxia-Inducing Factors

Hypoxia caused by blood stasis performs a key role in the development of oxidative stress and phlebitis. Under these conditions, several factors are expressed, including hypoxia-inducing factors (HIFs) or nuclear factor-κB (NF-κB)-dependent manners, and also matrix metalloproteinase inducers or activators, including the matrix metalloproteinase inducer. The expression of HIFs is associated with the regulation of further expression of genes involved in many processes, such as cellular metabolism, cell growth or death, cell proliferation, glycolysis, and immune response. Hypoxia signaling affects other cellular pathways, which include phosphoinositide 3-kinase (PI3K) signaling, the nuclear factor kappa-B (NF-κB) pathway, extracellular signal-regulated kinases (ERKs), and endoplasmic reticulum (ER) signaling and stress [69]. A key mediator of oxygen homeostasis is the transcription regulator hypoxia-inducible factor 1 (HIF-1). The presence of HIF-1 is required for the control of important physiological pathways. HIF-1α appears to perform a general role in regulating the transcription of all cells in response to hypoxia. So far, three HIF-α (HIF-1α, −2α and -3α) have been identified [70]. It is likely that HIF-1 performs a key role in the pathophysiology of cancer, cardiovascular disease, chronic lung and kidney disease, metabolic diseases, and reproductive diseases such as preeclampsia and endometriosis [71]. HIF-1α is expressed in most human tissues, possibly all [72]. In contrast, HIF-2α and HIF-3α appear in a small number of tissues (e.g., developing vascular endothelium and fetal lung) and have more limited or specialized roles in oxygen homeostasis [73,74]. In turn, HIF-1 induces the expression of matrix metalloproteinase-1, also known as fibroblast collagenase or interstitial collagenase. Matrix metalloproteinases may lead to inflammation and damage to endothelial cells and subsequent changes in the structure and function of the vein wall [75,76]. MMP leads to the degradation of the extracellular matrix, which in turn affects the reconstruction of the venous tissue, structural and degenerative changes in the vein wall, and damage to the valves. MMPs can cause changes in endothelial and venous smooth muscle function, leading to venous dilation. In addition, leukocyte infiltrates and inflammation of the veins promoting the release of additional MMPs, resulting in further dilatation of the venous wall and valve dysfunction and the development of chronic venous disease, including varicose veins [7].

### 4.2. The Role of Metalloproteinases in Varicose Veins

Blood stasis and hypertension in varicose veins leads to MMP expression and degradation of endothelial proteins [77,78,79]. Impairment of the endothelium promotes infiltration and activation of leukocytes, which causes an increase in elastase and lactoferrin [80,81]. Elevated plasma elastase levels in varicose veins do not depend on inflammatory complications (such as lipodermatosclerosis or ulceration) [81].

In turn, MMP-2 and MMP-9 are important factors that limit wound healing in venous ulcers. Elevated levels of MMP-9 in wound fluid and plasma have been observed [78,79]. As chronic wounds heal, a decrease in MMP-2 and MMP-9 is observed [82]. MMPs belong to Zn^2+^-dependent endopeptidases and can lead to the degradation of extracellular matrix (ECM) proteins. MMP activity in the vein is affected by the already mentioned inflammation, hydrostatic pressure, hypoxia, tissue metabolites. MMP expression leads to an increase in the proteolysis of various protein substrates in the extracellular matrix, mainly collagen and elastin, the consequence of which is the dysfunction of the venous wall. Metalloproteinases can also increase venous diameter by releasing endothelial-derived vasodilators and by activating potassium channels, resulting in hyperpolarization and relaxation of smooth muscle. In addition, metalloproteinases may also affect the proliferation, migration, differentiation, and apoptosis of vascular smooth muscle (VSM) cells. MMPs also affect the role of the endothelium and VSM contraction mechanisms [83,84]. On the other hand, the activity of MMPs is regulated by endogenous tissue inhibitors of metalloproteinases (TIMPs). Imbalance between MMPs and TIMPs may lead to venous dysfunction and the development of CVD [85]. In turn, in advanced clinical stages of varicose veins, reduced expression of MMP-9 and TGFβR3 genes was observed. On the other hand, expression of MMP-2 and TIMP-3 genes was increased. Decreased expression of TGFβR may be associated with a reduction in the participation of TGF-β1 in the disruption of the MMP/TIMP balance during the development of venous disease [76]. Moreover, matrix metalloproteases and their natural inhibitors are of key importance in the pathophysiology of acute and chronic thrombosis. Animal models of venous thrombosis have shown the effectiveness of anti-inflammatory treatment in removing thrombus and reducing damage to vessel walls [42].

### 4.3. Expression of Enzymes in Hypoxia

In cell cultures, it was shown that hypoxia led to the activation of leukocytes and endothelial cells, which released factors involved in the remodeling of the venous wall. Similar factors are involved in varicose veins [66]. In hypoxia, venous ECs produce greater amounts of HIF, a regulator of many genes involved in oxygen homeostasis. Under conditions of oxygen deficiency, VEGF and eNOS are expressed, and prostaglandin I2 and cyclooxygenase-2 (COX-2) increase. Moreover, in hypoxia, multinucleated cells (PMN) produce large amounts of superoxide anion radicals (O_2_^•−^) and leukotriene B 4, a molecule released in inflammation [66]. In addition, an increase in xanthine dehydrogenase (XD) and xanthine oxidase (XO) activity was observed in pulmonary artery endothelial (EC) cell cultures. However, no changes in the XD/XO ratio were observed. Hypoxia results in oxidative and inflammatory damage to the endothelium [86,87].

## 5. Oxidative Stress

Reactive oxygen species are produced in the body throughout our lives. They participate in transmitting signals within cells and between cells and in reactions that cause oxidative modifications in molecules and macromolecules. However, in a normally functioning body, there is a subtle balance between the release of ROS and their inactivation by specialized enzymatic and non-enzymatic systems present in mammalian cells and tissues. Excessive release of ROS and/or inefficient antioxidant systems leads to oxidative stress, which results in disturbances in the structure and function of cells and, consequently, in their degeneration and death. Oxidative damage accompanies many conditions and pathologies, as well as aging processes. Many studies have shown that reactive oxygen species perform an important role in the pathogenesis of varicose veins.

### 5.1. Reactive Oxygen Species and Antioxidants

ROS are produced by activated leukocytes, mainly neutrophils, which adhere to endothelial cells on the vascular wall [88,89,90,91]. There are various sources of ROS production, e.g., cytoplasmic membranes, endoplasmic reticulum, lysosomes, mitochondria, and peroxisomes [92,93]. The most efficient route of ROS production in the mitochondrial respiratory chain is electron transport, where 11 sites produce superoxide anion radical and/or hydrogen peroxide (H_2_O_2_) [94]. It is estimated that 1–3% of the oxygen flowing through the mitochondria is reduced to O_2_*^•^^−^* [95]. In addition to the superoxide previously mentioned, from which hydrogen peroxide is formed, other ROS include hydroxyl radicals (HO·), nitric oxide, singlet oxygen (^1^O_2_), and hypochlorous acid (HClO). Nitric oxide is a precursor to other reactive oxygen species often referred to as reactive nitrogen species such as nitrogen dioxide (NO_2_*^•^*), peroxynitrite (ONOO^−^), and others [96]. A frequently used indicator of neutrophil activation is the activity of myeloperoxidase, which participates in the oxidation of chlorides by hydrogen peroxide [97]. High myeloperoxidase activity was observed in varicose veins compared to the normal vein [88]. Moreover, activated neutrophils and monocytes release large amounts of proteases and phospholipases, enzymes, and reactive oxygen species, which are highly toxic to the vessel wall as well as the surrounding tissues [98]. It has also been shown that the interaction of activated neutrophils with the endothelium leads to the conversion of XD to XO in endothelial cells [99]. Xanthine oxidase (XO) is an important source of reactive oxygen species. ECs also have XO and can produce ROS [100]. ROS are also produced by NADPH oxidase (NOX) and nitric oxide synthase (NOS) [101,102]. Increased XO activity in the wall of varicose veins has been observed, especially in superficial thrombophlebitis. The expression of the enzyme concerned the lumen endothelium as well as the vasa vasorum [88]. Moreover, the blood of patients with VV showed a disturbed antioxidant defense mechanism, including reduced activity of superoxide dismutase (SOD) and total plasma antioxidant status, which consists of low molecular weight antioxidants such as glutathione (GSH), ascorbate (ASC), tocopherols (TOH), uric acid (UA), and thiols with low molecular weight. The decrease in antioxidant potential was correlated with the increase in oxidative stress expressed as a stress marker, malondialdehyde (MDA) [88,90]. Reduced SOD activity and total antioxidant status, as well as an increase in MDA levels, were also observed in the wall of a varicose vein compared to a normal vein [32,91]. In turn, in another study, an increase in SOD activity and antioxidant defense in the wall of varicose veins was observed [90]. Reactive oxygen species also caused damage to the subendothelial tissue, smooth muscle cell hyperplasia, and increased endothelial permeability [103]. It has been shown that the removal of a damaged vein (a varicose vein) led to a significant decrease in markers of oxidative stress in venous blood [66]. Additional amounts of reactive oxygen species are formed in hypoxia as a result of purine catabolism when hypoxanthine and xanthine are formed from ATP. Hypoxia and cytokines (TNF-α, IFN-γ, IL-6, and IL-1) lead to the activation of transcription of the xanthine oxidoreductase (XOR) gene [104,105]. XOR participates in purine catabolism. During the conversion of hypoxanthine to xanthine and xanthine to uric acid, catalyzed by xanthine dehydrogenase, two superoxide anion radical molecules are released [106]. In turn, the activity of xanthine oxidase (XO), in addition to purine catabolism, produces reactive oxygen species (ROS). A similar mechanism of action in the production of ROS is demonstrated by NADH oxidases [107]. In addition, the adhesion and activation of neutrophils and monocytes in the oxygen burst provide many other reactive oxygen species [108,109].

### 5.2. ROS and Enzymes in the Damage of Biological Material

ROS and proteolytic enzymes produced by active neutrophils and macrophages lead to the degradation of the proteins of the venous wall, collagen, and elastin. In addition, lipid peroxidation causes the disintegration of cell membranes and the release of additional amounts of proteases. Pathological veins contain more iron ions than normal ones. In addition, a higher concentration of zinc and copper ions was found in them [110]. Iron and copper ions perform a catalytic role in free radical reactions, contributing to the damage of biological material [111].

Oxidative stress in varicose veins is accompanied by inflammation. ROS perform a key role in endothelial damage, as well as in the development of atherosclerosis, inflammation, chronic venous insufficiency, and thrombosis (Figure 1). Chronic inflammation leads to the mobilization of certain chemical effectors and molecular signals in the cell, including reactive oxygen species and the enzymes COX-2, NAD(P)H oxidase, and inducible nitric oxide synthase (iNOS), which leads to increased production of ROS and nitric oxide [112,113]. The simultaneous production of NO· and superoxide (during phagocytosis) leads to the formation of peroxynitrite, which is a strong oxidizing and nitrating agent of tyrosine residues in proteins [96]. In turn, endothelial dysfunction is important in varicose veins and deep vein thrombosis (DVT) [114].

## 6. Hemoglobin as an Oxidant

Damage to the valves leads to slow blood flow or stagnation of blood in varicose veins leading to permanent hypoxia and inflammation in the vessel wall. Elevated concentrations of interleukin-6, fibrinogen, and hemoglobin (Hb) in the blood of varicose veins were found in comparison with the blood from the cubital vein of the same patient [115]. Moreover, high venous pressure and pathological capillaries characterized by increased permeability lead to the accumulation of fluid, the high molecular weight of molecules, and extravasated erythrocytes in the interstitial space [116]. The process of hemoglobin autoxidation increases in conditions of hypoxia, especially in microcirculation.

### 6.1. Antioxidant System in Red Blood Cells and Plasma

However, in red blood cells (RBCs), all oxidative reactions are suppressed by specialized antioxidant systems such as SOD, catalase (CAT), glutathione peroxidase (GPx), peroxiredoxin-2 (PRDX2), and low molecular weight antioxidants (GSH, ASC, TOH, and thiols). In turn, in the case of hemoglobin extravasated as a result of RBC hemolysis, oxidation reactions are more likely due to the less available antioxidant system. Therefore, the changed red blood cells undergo intravascular lysis under physiological conditions, releasing hemoglobin and its breakdown product, heme, into the plasma. Extravasated hemoglobin and heme are catalysts for free radical reactions [117]. To prevent the harmful effects of Hb and heme, they are bound by the acute phase plasma proteins, haptoglobin and hemopexin, respectively, and rapidly removed from the circulation [118].

### 6.2. Heme and Forms of Hemoglobin with Iron in Higher Oxidation States

It has been shown, in vitro, that free heme, in addition to generating ROS, can activate NF-κB-mediated expression of IL-1 and TNF-α, a 100-fold increase in transcription of pro-inflammatory genes and macrophage releases of cytokines such as IL-1 and TNF-α [119]. Moreover, heme is an independent activator of TLR4, initiating the secretion of TNF-α, ROS, and leukotriene B4 from macrophages and HMGB1 from testicular hepatocytes [120,121]. However, the oversaturation of this defense system leads to the formation of redox-active forms of Hb and heme in the circulation, which can initiate oxidative stress (Figure 2) [122]. Autooxidation of oxyhemoglobin (HbFe^2+^O_2_) leads to the release of the superoxide anion radical and methemoglobin (MetHbFe^3+^) [123,124]. The superoxide, in turn, dismutates or reacts with oxyhemoglobin to form hydrogen peroxide [125]. Unremoved hydrogen peroxide reacts with HbFe^2+^O_2_ to form the ferryl form of hemoglobin. In turn, the radical ferryl form (oxyferrylHb) is formed in the reaction of hydrogen peroxide with methemoglobin. Additionally, ferryl forms easily release heme and free Fe(III) ions in the presence of hydrogen peroxide [126]. Iron ions that are not bound by ferritin may, in the presence of reducing agents (e.g., ascorbic acid), form Fe(II) ions, which take part in the Fenton and or Haber–Weiss reactions by generating hydroxyl radicals or participate as catalysts in other free radical reactions. Moreover, heme and oxyferryl have pro-inflammatory properties, which increases their oxidative potential [127]. Both ferryl forms in which iron is in the 4th oxidation state, as well as hem, show high reactivity with biological material. These oxidizing agents are a potential source of oxidative reactions in plasma, but their oxidizing potential increases if lower molecular weight Hb dimers are absorbed into cells and tissues [126]. Hb is membrane-limited and is difficult to access by the cytosolic antioxidant systems of red blood cells. This process is intensified in hypoxic conditions because partially oxidized Hb has an increased rate of autoxidation and an even greater affinity for the red blood cell membrane [128]. Additionally, the oxidation of methemoglobin by hydrogen peroxide can lead to the formation of a highly reactive hydroxyl radical [108,129]. All these forms of ROS containing heme can cause damage to biological material including vascular endothelium [127].

## 7. Plasma and Red Blood Cells in CVeD

Red blood cells are the most abundant cells present in the blood. In addition to oxygen transport, RBCs are also important modulators of the innate immune response, binding and removing chemokines, nucleic acids, and pathogens from the circulation. They perform important functions in innate and specific immune reactions in controlling inflammatory processes. For example, the main binding site for chemokines is the DARC antigen receptor. RBCs have been shown to take up CXCL8, inactivating the CXCL8-dependent chemokine gradient, which blocks neutrophil recruitment [130]. DARC also regulates with high affinity the chemokines CXC and CC in addition to CXCL8 (and other immunomodulatory proteins [131]. The binding of inflammatory molecules by RBC receptors leads to impaired immune response by inhibiting neutrophil signaling [132]. This slowing mechanism could be important because excessive activation of the immune system may cause excessive inflammation and tissue damage. When threatened, this system can promote activation of the immune system [133]. RBCs effectively maintain homeostasis by capturing chemokines from the inflammatory region and releasing them when their concentrations decrease [134].

### 7.1. mtDNA in Varicose Veins

In CVeD, the innate immunity of RBCs is impaired, and an insufficient inflammatory response may cause damage to the muscle and microvascular tissue of the lower limb [135]. Red blood cells bind not only chemokines, but RBCs have also been shown to bind molecules that may be involved in inflammatory responses, such as nucleic acids, including mitochondria-derived mtDNA. Mitochondria are an endogenous reservoir of inflammatory nucleic acids and can be released into the circulation by stimulated immune cells. mtDNA contains CpG motifs and can induce pro-inflammatory signaling by binding to Toll-like receptor 9 (TLR9) DNA regions through CpG-containing motifs [136]. One recent study showed a decrease in the level of total mtDNA in samples from varicose vein samples compared to samples from non-varicose vein tissue samples. Moreover, a lower percentage of mtDNA without deletions and a decrease in the relative mitochondrial membrane potential in ECs were observed in VV. These results demonstrate the involvement of mtDNA in the pathogenesis of VV and suggest a possible reduction in mitochondrial function in this disease [137].

### 7.2. Alterations in Red Blood Cells Properties in CVeD

When examining the rheological properties of red blood cells in patients with CVeD, higher values related to RBC elongation and susceptibility to aggregation were shown compared to controls. The authors suggest that the increase in RBC deformability may be associated with greater resistance to flow in microcirculation [138]. Another study showed that red blood cells originating from varicose veins and antecubital veins of patients with CVeD had different rheological properties, i.e., deformability and aggregation. RBCs originating from varicose veins were more deformable and more susceptible to aggregation than RBCs from the antecubital vein [139]. The higher deformability of RBCs from varicose veins may be due to increased resistance to microcirculation. In summary, this study demonstrates that heterogeneity in red blood cell rheology is influenced by both age and cardiovascular disease. Further studies of red blood cells in CVeD have shown that changes in RBC rheology occur among both young and old red blood cell subpopulations. It was confirmed that RBC subpopulations of CVeD patients were characterized by higher plasticity leading to changes in shape and greater susceptibility to aggregate formation and stability than the corresponding control subpopulations [140]. Horecka and colleagues found a decrease in superoxide dismutase activity in erythrocytes from patients with varicose veins compared to a group of healthy people [90]. However, no changes were observed in the concentration of glutathione in plasma and varicose veins. Nonetheless, there was a decrease in the concentration of total antioxidant status (TAS) in the plasma and wall of varicose veins compared to the control group.

We compared changes in the properties of red cells derived from varicose veins with RBCs taken from the cubital vein of the same patients with chronic venous disease. The studies were carried out on whole red blood cells using the spin labeling method in EPR spectroscopy. The alterations in the structure of endothelial cells were accompanied by changes in red blood cells isolated from the pathological vein. A decrease in the internal viscosity of erythrocytes from varicose veins was observed. In red blood cells derived from varicose veins, we showed a significant decrease in lipid fluidity in the subsurface area of the membrane. In addition, in varicose vein membranes, we demonstrated the structure of the membrane cytoskeleton (structural stiffening) in comparison with RBCs obtained from the ulnar vein. These changes were correlated with a decrease in the intrinsic viscosity of RBCs and an increase in the osmotic fragility of RBCs from varicose veins, and thus greater sensitivity to hemolysis than red cells from a peripheral vein [141]. An increase in the stiffness of red blood cell membranes and changes in the viscosity of the cell interior lead to less deformability of the cell. It has been shown that morphologically changed red blood cells are more susceptible to hemolysis [142,143]. These results may indicate changes in the blood hemorrhage of varicose veins but they also indicate the possibility of releasing hemoglobin from RBCs, a catalyst for free radical reactions, and, consequently, the development of oxidative stress in varicose veins.

We demonstrated that hemoglobin modified by ROS or bound to the plasma membrane is present in erythrocytes from the varicose vein [144]. It has been shown that irreversible oxidation of this cysteine residue, βCys93, in the globin chain may lead to the breakdown of the Hb structure and, consequently, the release of heme [145]. βCys93 acts as a nitric oxide carrier and is also necessary for proper tissue oxygenation and proper functioning of the circulatory system [146]. It was found that the higher the degree of hemoglobin oxidation, Fe(III) to Fe(V), the greater the binding of Hb to the membrane [147]. Moreover, approximately 50% of this bond was associated with the participation of reactive thiol groups. Membrane-bound hemoglobin has been shown to form a high molecular weight complex with cytoskeletal proteins such as spectrin, ankyrin, and band 4.2. This complex was active in the redox process and initiated lipid peroxidation, which may consequently cause cell damage [147]. A modified form of hemoglobin has been observed in erythrocytes from patients with chronic kidney disease, where oxidative stress plays a key role in the damage of plasma and its components [148].

### 7.3. Antioxidants and Oxidative Stress in Plasma and Red Blood Cells in CVeD

To demonstrate the role of oxidative stress in the blood of varicose veins, we compared markers of oxidative stress in plasma and red blood cells taken from varicose veins and the cubital vein of the same patients. In plasma from varicose veins, we found lipid and protein peroxidation products, as indicated by higher levels of a thiobarbituric acid reactive substance (TBARS) and protein carbonyl compounds, as well as by a decrease in thiol compounds [144]. Glutathione and thiols present in red blood cells perform an important role in protecting lipids, proteins, and other macromolecules against oxidative damage. Thiols are oxidized by mild oxidants such as superoxide and hydrogen peroxide. For this reason, they perform an important role in protecting cells against ROS in vivo [149,150]. In addition, we demonstrated that treatment of red blood cells with diosmin and bromelain stimulated the biosynthesis of glutathione and thiols in RBCs. This mechanism may be important in the action of these substances in varicose veins, because, apart from other properties, these compounds contribute to the reduction of oxidative stress [57]. An increase in the level of oxidative stress parameters in plasma, e.g., TBARS, and protein carbonyls, as well as decreased catalase activity and thiol levels, were also observed in a group of patients with varicose veins compared to a group of healthy volunteers [151,152]. Oxidative damage to proteins and lipids was accompanied by a decrease in the level of thiols, a decrease in the level of non-enzymatic antioxidant capacity (NEAC), as well as a decrease in catalase, and acetylcholinesterase (AChE) activities in plasma derived from varicose veins [144]. It has been shown that the decrease in CAT activity can be initiated by hydroxyl radicals, superoxide, and H_2_O_2_, but not by organic peroxides [153,154]. Increased oxidative stress in varicose veins may be the result of both excessive ROS production and the failure of antioxidant systems [144]. The observed changes in the structure of red blood cells and endothelial cells derived from VV may also reflect changes occurring in other cells derived from the pathological vein. Higher levels of protein S, vWf, VEGF, and IL-12 proteins were found in the plasma of patients with varicose veins compared to the control group, while the levels of protein C, fibrinogen, homocysteine, and PGI2 did not show significant differences [155]. In addition, the adhesion of leukocytes to the vascular wall may initiate additional oxidative stress as well as affect the remodeling of the vascular wall.

## 8. Varicose Veins Treatment

In addition to invasive/surgical methods such as sclerotherapy, endovascular laser or radiofrequency ablation, microphlebectomy [156,157], treatment with phlebotonic drugs is used. Venoactive drugs of different mechanisms of action in many cases are effective and safe way of treatment for patients with CVeD (Table 1). Most often, these are drugs of natural origin obtained by extraction from plants [114].

### 8.1. Flavonoid Derivatives

Flavonoids belong to the group of natural substances widely found in nature and can be found in fruits, vegetables, grains, bark, roots, stems, flowers, tea, and wine [158]. Flavonoids found in the plant world are based on the chromane ring (benzodihydropyran). Their diversity is related to the presence of a double bond in the pyran part of the ring and the position of the carbonyl group (α- and γ-pyrone). Therefore, flavonoids can be divided into various groups, such as chalcones (the precursors of all flavonoids), flavans, isoflavans, neoflavans, flavonolignans, aurones, leucoanthocyanidins, and anthocyanidins (containing the flavylium ion) [114]. Additionally, the presence of hydroxyl groups in various positions of the chromate ring promotes the formation of glycosides. Table 1 below shows the various flavonoids most commonly used in CVeD therapy.

Among the numerous derivatives of anthocyanidins containing the flavylium ion, some substances are relatively simple in structure, such as pelargonidin, cyanidin, delphinidin, petunidin, peonidin, and malvidin, which has a protective effect in cardiovascular diseases. In turn, anthocyanins, which are glycosides of anthocyanidins, have a coloring function and are found in berries, e.g., wild blueberries, bilberries, cranberries, elderberries, and strawberries. In turn, multi-colored anthocyanins, which are anthocyanidin glycosides, are found in berries including, among others, wild blueberries, blueberries, cranberries, elderberries, and strawberries. Procyanidins, which are oligomers of catechin and epicatechin have a more complex structure. They are found in cranberries as cranberry tannins but also contain other procyanidins, myricetin derivatives, and quercetin, as well as other derivatives. [159].

A drug widely used in CVeD therapy is diosmin, which is a semisynthetic flavonoid [160]. The drug is available in the form of a micronized fraction as MPFF (Daflon) and contains mainly diosmin (90%) and other active flavonoids, such as hesperidin, diosmetin, linarin and isoorhoifolin from *Rutaceae aurantiae* [161]. Diosmin is characterized by a broad spectrum of action in vessels, as by increasing venous tone and lymphatic flow, it affects the elasticity of vessels [162,163]. It has anti-swelling properties, reduces the permeability of blood vessel walls, and improves blood circulation in capillaries. Like other flavonoids, it has anti-inflammatory and antioxidant properties and has positive effects on the elasticity of the vessels [164,165]. Moreover, it inhibits leukocyte adhesion, activation, and aggregation of platelets and the complement system [166,167]. It has been shown that diosmin decreases COX-1 activity [167].

Rutin is a flavonoid glycoside found in citrus fruits and many plants, including buckwheat, leaves, and petioles of Rheum species and asparagus. In addition, it is a flavonol found in large amounts in peaches and green tea. However, the largest quantities of rutin can be found in young caper leaves (*Capparis spinosa* L.) [168]. Rutin has anti-inflammatory and antioxidant properties. It inhibits pro-inflammatory signaling of VCAM-1, ICAM-1, and E-selectin. On the one hand, it inhibits PAF, on the other hand, it induces NOS, increasing the release of NO. It inhibits adhesion and migration of leucocytes to inflamed endothelium as well as neutrophils adhesion and migration. It reduces the expression of NF-kB and of the pro-inflammatory cytokines IL-6 and TN-a. It has antihypertensive properties.

Troxerutin is a rutoside (a derivative of rutin), a flavonoid and flavonol naturally occurring in citrus fruits, *Sophora japonica*, and flowers of the *Styphnolobium japonicum* tree. It is also found in many other plants such as tea, coffee, and cereal grains as well as vegetables. Troxerutin is a derivative of the naturally occurring rutin. It has antioxidant, anti-inflammatory, neuroprotective, renoprotective, and antidiabetic properties [169]. In addition, this rutoside improves the rheological properties of blood, inhibits the aggregation of erythrocytes and platelets, improves the deformability and aggregation of erythrocytes as well as plasma viscosity, which is important for microcirculation, including microcirculation in the retina.

Troxerutin has an antithrombotic effect, and also increases the tension of the walls of venous vessels and regulates their permeability. It is used to treat venous and lymphatic circulation disorders, especially in the lower limbs, phlebitis, post-thrombotic syndrome, and varicose veins of the lower limbs and anus [170,171]. Its properties of reducing capillary permeability have also been used in the treatment of diabetic retinopathy, as well as in vascular damage to the retina and in the treatment of blood clots and subconjunctival hemorrhages [172].

Oxerutins are semi-synthetic derivatives of rutin and contain a standardized mixture of 5% mono-, 34% di-, 46% tri- and 5% O-β-hydroxyethyl tetrarutosides and are obtained from the *Sophora japonica* plant [173]. Oxerutins are used in the treatment of chronic venous disease [174]. Hydroxyethylrutosides have a protective effect on the vascular endothelium and also have a strong affinity for the venous wall [166]. Pharmacological and clinical studies have shown that oxerutins have a positive effect on abnormalities related to capillary permeability and also have an impact on red blood cell membranes, their deformation, and aggregation [175,176]. Moreover, they have anti-edema effects and inhibit the synthesis of prostaglandins [177]. Oxerutins inhibit the recruitment and activation of neutrophils by stimulating the endothelium during blood stasis [178]. It has been shown that the most important pharmacological effect of oxerutins is the inhibition of microvascular permeability and the reduction of edema [179].

Hesperidin is a flavonoid found in high concentrations in citrus fruits. It has a beneficial effect on the cardiovascular system and type II diabetes. It has anti-inflammatory, antioxidant, and antimicrobial properties. The anti-inflammatory effect of Hesperidin is associated with the inhibition of the p38 MAPK signaling pathway, which can significantly reduce the expression of pro-inflammatory cytokines IL-1β and IL-6, IL-8, IL-18, and TNF-α, both in cultured macrophages and in mice [180]. The protective effect of hesperidin on the cardiovascular system is associated with lowering blood pressure and glucose levels. Moreover, this flavonoid reduces platelet aggregation and increases the expression of antioxidant enzymes CAT and SOD [181]. Hesperidin is used in cardiovascular diseases (i.e., high blood pressure, stroke, and heart attack), as well as in blood vessel diseases (venous ulcers and hemorrhoids). Recently, it has been shown that hesperidin can inhibit transcription factors and control regulatory enzymes of inflammatory mediators such as nuclear factor-kappa B (NF-κB), iNOS and COX-2. Moreover, hesperidin improves cellular antioxidant defense by activating the ERK/Nrf2 signaling pathway [182].

Traditional medicine uses extracts of plants, including pine bark. Pycnogenol (PG) is a proprietary, standardized extract from the bark of French maritime pine (*Pinus pinaster* ssp. *atlantica*), containing various substances with antioxidant, anti-inflammatory, and antiplatelet properties. Pycnogenol contains a large amount of proanthocyanidins (65–75%) as well as caffeic acid, catechin, ferulic acid, and taxifolin [183]. Its action is also related to its vasodilatory and antithrombotic effect. PG is used to treat chronic venous insufficiency and related venous disorders, including deep vein thrombosis, post-thrombotic syndrome, venous ulcers, and acute hemorrhoids. It was shown that Pycnogenol improves endothelial function in CAD patients by reducing oxidative stress [184]. It has been shown that PG increases the resistance of capillaries and improves blood flow, increases the release of NO from vascular endothelial cells, and protects the endothelium against the damaging effects of ROS [185]. Moreover, PG regenerates the ascorbyl radical and has a protective role against endogenous vitamin E and glutathione in the cellular antioxidant system [183,186]. PG inhibits the nuclear factor (NF-κB), and modulates NO metabolism by scavenging NO and inhibiting iNOS mRNA expression and iNOS activity. The action of PG is also related to gene inhibition expression of pro-inflammatory cytokines (IL-1 and IL-2) and inhibition of the activity of both cyclooxygenases (COX-1 and COX-2) [187]. It also has a vasodilating and anticoagulant effect and stabilizes collagen [188]. Pycnogenol is used in vein diseases including varicose veins.

### 8.2. Coumarin Derivatives

Coumarins are derivatives of 2H-chromen-2-one, occurring in tonka beans (*Dipteryx odorata* Wild), but also in other plants such as *Rutaceae*, *Umbelliferae*, *Clusiaceae*, *Guttiferae*, and others. Products of natural origin, coumarin derivatives, have broad pharmacological and therapeutic properties, which include anti-inflammatory, antioxidant, antiviral, antibacterial, anti-coagulant, anti-edema, and anti-cancer. For example, coumarins have shown anti-inflammatory properties; however, they have limited use as pharmaceuticals such as in the treatment of lymphedema due to their ability to increase plasma antithrombin levels [189,190]. However, synthetic coumarin derivatives, such as warfarin, acenocoumarol, fraxetin, fraxin, esculetin, and aesculin, have been used as drugs. They are used as oral anticoagulants to treat deep vein thrombosis and pulmonary embolism, as well as to prevent stroke in people suffering from atrial fibrillation and valvular heart disease or with artificial heart valves [191]. In turn, esculetin is a derivative of coumarin, found in various plants, such as *Aesculus hippocastanum* L., *Sonchus grandifolius*, *Aesculus turbinate*, and others. Esculetin inhibits the synthesis of leukotrienes B4, activation of the NF-κB and MPAK pathways, and the production of inflammatory cytokines. In turn, lipid peroxidation, caused by increased expression via TGF-β and decreased activity of antioxidant enzymes, is inhibited by esculetin. Additionally, esculin (esculetin-6-glucoside) has protective properties on the walls of capillary vessels [192]. Esculetin has been shown to scavenge free radicals produced during lipid peroxidation and to also lead to an increase in the levels of antioxidant enzymes such as CAT, SOD, and GPx [193]. In addition, esculetin reduces the expression of inflammatory cytokines and chemokines such as TNF-α, IL-1β, IL-6, CCL2 and iNOS [192].

### 8.3. Saponins Derivatives

Aescin, found in horse chestnut seeds (*Aesculus hippocastanum* L.) and extracts from *Centella Asiatica* and *Ruscus aculeatus* (Butcher’s broom), are used in the treatment of chronic venous diseases. Aescin is a mixture of two triterpene saponins, α and β isomers [194]. The active isomer is form β. Aescin has anti-edematous, anti-inflammatory, and venotonic effects, which may be related to reducing vascular permeability and inhibiting the negative effects of hypoxia [166]. Aescin is effective in vascular insufficiency because, in addition to its anti-edematous and vascular protective effect, it also has diuretic properties. This saponin has a toning effect on vessels by sealing and strengthening the capillary walls, while also having anti-exudation and anti-edema effects [194]. Additionally, horse chestnut extracts contain flavonol glycosides, mono-, di-, and tri-sugar derivatives of quercetin and kaempferol, leucoanthocyanins and procyanidins, as well as tannins. Aescin, in the form of dragees and as a transdermal gel, has proven effective in the treatment of chronic venous insufficiency. In addition to its anti-inflammatory properties, aescin reduces vascular permeability in inflamed tissues, which leads to inhibition of edema. Furthermore, aescin prevents disruption of the normal expression and distribution of the platelet endothelial cell adhesion molecule-1 in hypoxia, which may help explain its protective effect on vascular permeability [194]. Aescin induces iNOS, which increases the permeability of endothelial cells to calcium ions. Moreover, it induces the release of prostaglandin F2α.

Other natural substances used in chronic therapy of CVeD, especially in chronic vascular insufficiency, include steroid saponins (ruscosides) isolated from butcher’s broom (*Ruscus aculeatus)* and gotu cola (*Centella asiatica*). In addition to steroid saponins, *Ruscus aculeatus* contains other substances such as aglycones (ruscogenin and neoruscogenin), as well as benzofurans, e.g., euparone, furostanols and flavonoids [195]. The best-studied therapeutic properties are displayed by *R. aculeatus* as a very old phlebotherapeutic agent. *R. aculeatus* L. was popular in Germany and France for the treatment of chronic venous insufficiency, varicose veins, hemorrhoids, and orthostatic hypotension [196]. Sapogenins obtained from *R. aculeatus* showed anti-inflammatory effects. In patients with chronic venous insufficiency treated with Ruscus extract, constant venous tension, and improved venous emptying were observed, where the latter was not observed in placebo-treated patients [197]. Butcher’s broom extract was shown to inhibit endothelial cell activation due to hypoxia, a condition occurring in venous blood stasis. This effect was associated with a decrease in ATP concentration and activation of phospholipase A2, and subsequent increase in neutrophil adhesion. These observations explain the beneficial therapeutic effects of the extract in the treatment of chronic venous disease [198]. Ruscogenin has been shown to have significant anti-inflammatory and antithrombotic effects. It inhibits the adhesion of leukocytes to the human umbilical vein endothelial cell line damaged by TNF-α. Ruscogenin significantly inhibits leukocyte migration into the peritoneum in mice in a dose-dependent manner induced by zymosan. Ruscogenin also inhibits TNF-α-induced ICAM-1 overexpression at the mRNA and protein levels. Moreover, it significantly inhibits NF-κB activation, reducing NF-κB p65 translocation and DNA binding activity [199].

### 8.4. Other Natural Bioactive Compounds

Another extract with anti-inflammatory, antioxidant, and venoactive properties is Ginkgo biloba (Gb) extract, which contains flavonoid glycosides, terpenes, lactones, proanthocyanidins, carboxylic acids, and high molecular weight compounds, as well as sterols, carotenoids, polyprenols, long-chain hydrocarbons [200]. Compounds with interesting properties are diterpenes with a cage skeleton, such as ginkgolides A, B, C, J, and M, and sequiterpenoids (bilobalide) [201,202]. Gb inhibits the expression of ICAM-1 and VCAM-1, which promote the recruitment of leukocytes, platelets, and erythrocytes, as well as their adhesion to the vein wall [203]. Gb also performs an important role in peripheral circulation and microcirculation. Ginkgolide B (GB) is the most potent antagonist of platelet-activating factor (PAF) and an intracellular mediator involved in the process of platelet aggregation and inflammatory reactions [204]. Ginkgolide A and B have antioxidant properties, inhibit the formation of ROS and induce an increase in SOD activity [205,206]. It has been shown that ginkgolides B, C, J, M, and bilobalide react with superoxide and its protonated (HO_2_*^•^*) form [207]. Ginkgolide B has anti-ischemic and antioxidant effects and anticonvulsants and control inflammation. It is used to treat thrombosis in clinical practice [208]. GB inhibits PLA_2_ activity and the production of TNF-α.

### 8.5. Synthetic Drugs

Although there are many synthetic drugs, such as naftazone, calcium dobesilate, benzarone, and others, drugs of natural origin are more widely used in therapy. For example, naphthazone was expected to alleviate lower limb edema, but there was no difference between the drug’s use in VV patients and placebo, although previous studies showed that it accelerated human saphenous vein endothelial proliferation in vitro and improved endothelial cellular metabolism in hypoxic conditions [209]. In turn, research presented in 2004 showed that calcium dobesilate, which increases lymphatic drainage and can increase venous tone, significantly improved the condition of patients in terms of edema. Whereas, research conducted in 2009 did not show any differences between the research group and placebo after three months of treatment [210]. However, greater hopes for the treatment of VV are associated with venotonic drugs, sulodexide, and other endogenous and exogenous MMP inhibitors. The goal of these drugs is to reduce the progression of CVeD and VV. For example, sulodexide has shown benefits in VLU and CVeD, which is associated with improved venous smooth muscle contraction. A separate group includes drugs that induce endogenous inhibitors of tissue metalloproteinases and the use of synthetic MMP inhibitors, providing other methods of treating CVeD [211].

## 9. Conclusions

Many of the cited studies have shown that patients with varicose veins are exposed to chronic inflammation and oxidative stress. In addition to excessive production of ROS in the blood of the patients, impairment of the antioxidant defense mechanism is observed. Taking into account the anti-inflammatory and antioxidant properties, lowering blood pressure, inhibition of platelets, improving endothelial homeostasis, increasing blood perfusion, and others, with no side effects, flavonoids, saponins, plant extracts (Pycnogenol, *Ruscus aculeatus*, *Centella asiatica* and *Gingko biloba*) should find wider use in CVeD therapy.

## Figures and Tables

**Figure 1 ijms-25-01560-f001:**
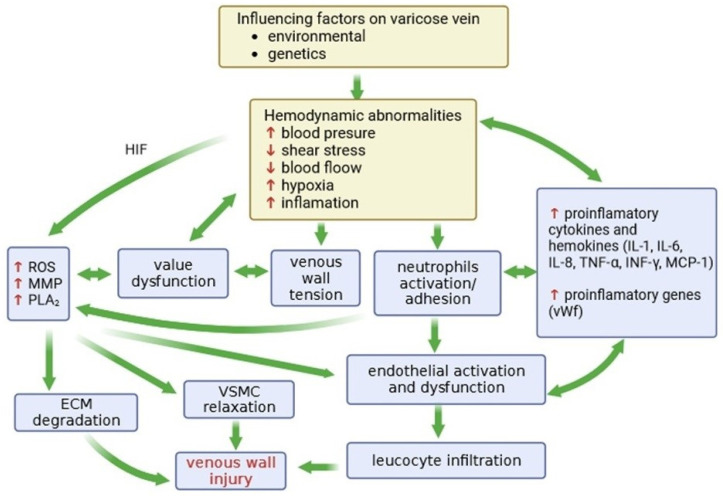
Factors influencing venous wall remodeling. The diagram shows the interplay of mechanisms involved in the pathophysiology of vein damage in CVeD, starting with the environmental and genetic factors that contribute to hemodynamic abnormalities. As a consequence, this leads to an increase in blood pressure and a slowdown in blood flow, shear stress, hypoxia and the development of inflammation, in which metalloproteinases (MMPs) degrading the extracellular matrix (ECM), reactive oxygen species (ROS) inducing oxidative damage, and phospholipase (PLA_2_) that degrades membrane lipids, take part. Moreover, pro-inflammatory cytokines are involved in the development of CVeD: interleukins (IL-1, -6, -8), TNF-α (tumor necrosis factor), IFN-γ (interferon gamma), MCP-1 (monocyte chemoattractant protein 1), and vWf (von Willebrand factor). ↑—increase, ↓—decrease.

**Figure 2 ijms-25-01560-f002:**
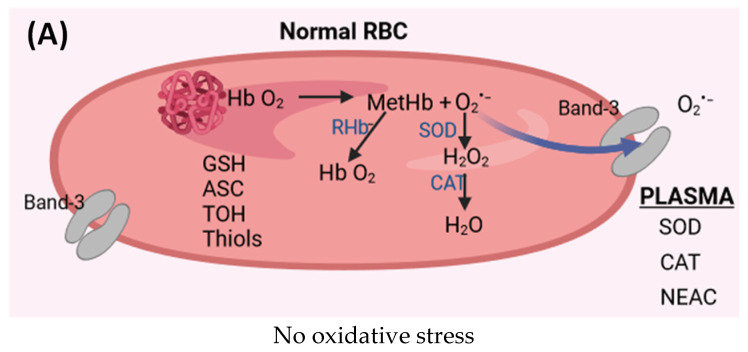
(**A**) Oxidation of hemoglobin and antioxidant systems in normal red blood cells (RBCs). RBCs are characterized by high resistance to hemolysis. Normal activity of antioxidant enzymes (SOD, CAT) and low molecular weight antioxidants (GSH, ASC, TOH, thiols) in RBC and plasma. (**B**) Altered red blood cells in the varicose vein (VV) during blood stasis and hypoxia. The activity of antioxidant enzymes (SOD, CAT) is lower as are GSH, ASC, TOH, and thiols levels. An overproduction of ROS may appear under these conditions, which leads to oxidative stress inside RBCs, and an altered hemoglobin binds to the cell membrane. ROS can also diffuse through the RBC membrane into the plasma. Reduced activity of antioxidant enzymes (SOD, CAT) and non-enzymatic antioxidant capacity (NEAC) in the plasma was observed, which leads to oxidative stress. RBC from varicose veins has a low resistance to hemolysis, which leads to the release of hemoglobin from the RBC into the plasma. Additionally, reduced activity of antioxidant enzymes and NEAC in plasma leads to oxidative stress. Possible reactions of extracellular hemoglobin and heme are also presented, which lead to the formation of hemoglobin forms with a higher oxidation level (ferryl form and radical ferryl form). Moreover, there is a release of heme from hemoglobin and the decomposition of heme with the release of free iron II ions in the plasma, which participate in Fenton reaction generating a hydroxyl radical. Severe oxidative stress occurs under these conditions. Abbreviation: HbO_2_ (oxyhemoglobin); MetHb (methemoglobin); SOD (superoxide dismutase); CAT (catalase); RHb (hemoglobin reductase); GSH (glutathione); ASC (ascorbate); TOH (tocopherols); NEAC (non-enzymatic antioxidant capacity); (Hb^4+^O) ferryl form of hemoglobin; and (**^•^**Hb^4+^O) the radical ferryl form of hemoglobin. ↑ —increase, ↓—decrease.

**Table 1 ijms-25-01560-t001:** Classification of natural venoactive drugs.

Group	Substance	Pharmacological Action
Flavonoids		
Diosmin	7-disaccharide derivative of diosmetin. The drug is available in the form of micronized purified flavonoid fraction (MPFF, Daflon-90% and 10% other active flavonoids: hesperidin, diosmetin, linarin and isoorhoifolin from *Rutaceae aurantiae*.	Diosmin acts on vessels, increasing venous tone, lymphatic flow, and improving vascular elasticity. It also has an anti-edema effect, reduces the permeability of blood vessel walls, and improves blood circulation in the capillaries. It has anti-inflammatory and antioxidant properties and has positive effects on the elasticity of the vessels. It inhibits leukocyte adhesion, activation of platelets and the complement system, and decreases the COX-1 activity.
Rutin	3-disaccharide derivative of quercetin found in citrus fruits, buckwheat, asparagus, peaches, green tea, and capers.	It has anti-inflammatory and antioxidant properties. It inhibits pro-inflammatory signaling of VCAM-1, ICAM-1, and E-selectin. It inhibits PAF but induces NOS, increasing NO release. It inhibits adhesion and migration of leucocyte to inflamed endothelium and neutrophils adhesion and migration. It reduces the expression of NF-κB and of the pro-inflammatory cytokines IL-6 and TN-α. It has antihypertensive properties.
Troxerutin	3-rutoside, disaccharide derivative. It occurs in citrus fruits, tea, coffee cereal grains, and vegetables.	It improves the rheological properties of blood, inhibits the aggregation of erythrocytes and platelets, improves the deformability and aggregation of erythrocytes, as well as the viscosity of plasma and microcirculation in the retina. It has antithrombotic properties, increases the tension of venous walls, and regulates their permeability. It is used in the treatment of venous and lymphatic circulation disorders, especially in the lower limbs, phlebitis, post-thrombotic syndrome, varicose veins of the lower limbs and anus.
Oxerutin	3-disaccharide derivative hydroxyethylrutoside. Oxerutines are semi-synthetic derivatives of rutin. The standardized mixture used as a pharmacological preparation contains 5% mono-, 34% di-, 46% tri-, and 5% O-b-hydroxyethyl tetrarutosides. Oxerutins are obtained from the *Sophora japonica* plant.	Oxerutins are used in the treatment of chronic venous disease. Hydroxyethylrutosides have a protective effect on the vascular endothelium and also have a strong affinity for the venous wall. They inhibit the permeability of microvessels and reduce swelling. They have a positive effect on red blood cell membranes, their deformation and aggregation, and improve the rheological properties of RBCs. In addition to their anti-edema effect, they inhibit the synthesis of prostaglandins. They inhibit the recruitment and activation of neutrophils by stimulating the endothelium during blood stasis.
Hesperidin	7-disaccharide derivative of flavanon. It occurs in large concentrations in citrus fruits.	It has anti-inflammatory, antioxidant, and antimicrobial properties. The anti-inflammatory effect of Hesperidin was associated with the inhibition of the p38 MAPK signaling pathway, and the expression of pro-inflammatory cytokines. This flavonoid reduces platelet aggregation and increases the expression of antioxidant enzymes CAT and SOD. Hesperidin can inhibit inflammatory mediators, NF-κB, iNOS, and COX-2, and activate the ERK/Nrf2 signaling pathway, improving cellular antioxidant defense.
Pycnogenol	Extracts from pine tree bark (*Pinus pinaster* ssp. *atlantica)* containing many flavonoids, 65–75% of proanthocyanidins, and phenolic acid.	Pycnogenol exerts antioxidative, anti-inflammatory, and anti-platelet effects. It increases the resistance of capillaries and improves blood flow, increases the release of NO from vascular endothelial cells and protects the endothelium against the damaging effects of ROS. It regenerates the ascorbyl radical and has a protective role against endogenous a-tokoferol and GSH in the cellular antioxidant system. It inhibits nuclear factor (NF-κB), and modulates NO metabolism by scavenging NO and inhibiting iNOS mRNA expression and iNOS activity. PG inhibits the gene responsible for the expression of pro-inflammatory cytokines (IL-1 and IL-2) and inhibits the activities of COX-1 and COX-2. It dilates blood vessels, has antithrombotic properties, and stabilizes collagen.
Coumarin	chromen-2-on derivatives. They occur in Melilot (*Melilotus officinalis*) and Woodruff (*Asperula odorata*). Synthetic derivatives: acenocoumarol, warfarin, fraxetin, fraxin, esculetin.	Coumarin derivatives, have broad pharmacological and therapeutic properties, such as anti-inflammatory, antioxidant, antiviral, antibacterial, anti-coagulant, anti-edema, and anti-cancer effects. Esculetin coumarin derivatives scavenge free radicals generated during lipid peroxidation, improving the levels of antioxidant enzymes such as CAT, SOD, and GPX. Esculetin inhibits the synthesis of leukotrienes B4, activation of the NF-κB and MPAK pathways and the expression of inflammatory cytokines and chemokines, such as TNF-α, IL-1β, IL-6, CCL2, and iNOS.
Saponins		
Aescin	Horse chestnut seed extracts (*Aesculus hippocastanum* L.)	Aescin has anti-inflammatory properties, reduces vascular permeability in inflamed tissues, which leads to inhibition of edema. This saponin prevents disruption of the normal expression and distribution of the platelet endothelial cell adhesion molecule in hypoxia, which may help explain its protective effect on vascular permeability. Aescin induces iNOS, which increases the permeability of EC to calcium ions, and induces the release of prostaglandin F2α. Aescin reduces the content of TNF-α and IL-1β.
Steroid saponins	*Ruscus aculeatus* (Butcher’s broom) and *Centella asiatica* extracts	The extract inhibits endothelial cell activation due to hypoxia, a condition occurring in venous blood stasis. This effect is associated with a decrease in ATP concentration and activation of phospholipase A2, and a subsequent increase in neutrophil adhesion. These observations explain the beneficial therapeutic effects of the extract in the treatment of chronic venous disease. Ruscogenin has strong anti-inflammatory and antithrombotic effects. Ruscogenin significantly inhibits leukocyte migration into the peritoneum. Ruscogenin also inhibits TNF-α-induced ICAM-1 overexpression at the mRNA and protein levels. It significantly inhibits NF-κB activation, reducing NF-κB p65 translocation and DNA binding activity.
Ginkgo biloba (Gb) extract	Extracts of Ginkgo biloba contain flavonoid glycosides, terpenes, lactones, proanthocyanidins, carboxylic acids, and high molecular weight compounds, as well as sterols, carotenoids, polyphenols, and long-chain hydrocarbons. Compounds with interesting properties are diterpenes with a caged skeleton, such as ginkgolides A, B, C, J, and M, and sesquiterpenoids (bilobalide)	Gb inhibits the expression of ICAM1 and VCAM, and also performs an important role in peripheral circulation and microcirculation. Ginkgolide B (GB) is the most potent antagonist of platelet-activating factor (PAF) and an intracellular mediator involved in the process of platelet aggregation and inflammatory reactions. Ginkgolide A and B have antioxidant properties, they inhibit the formation of free radicals, and protect against OS. Ginkgolides B, C, J, M, and bilobalide react with superoxide and its protonated form. Ginkgolide B has anti-ischemic and antioxidant effects, and it is an anticonvulsant and controls inflammation. It is used to treat thrombosis in clinical practice. GB inhibits PLA_2_ activity and the production of TNF-α.

## Data Availability

No new data were created or analyzed in this study. Data sharing is not applicable to this article.

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
