# Peer review of "Factors Influencing Venous Remodeling in the Development of Varicose Veins of the Lower Limbs"

_ijms, 2024, doi:10.3390/ijms25031560_

Round 1

Reviewer 1 Report

Comments and Suggestions for Authors

A very thorough paper although I find that some of the concepts are somewhat repetitive. Overall, the paper is very difficult to read and can be further improved with better paragraphing and keeping the points concise. It appears that the draft is at least 9500-10000 words. I would suggest that a draft half this length should be sufficient to bring forward the key points.

Other comments:

Figure on page 1 – please provide abbreviations

CVD is typically an acronym used for cardiovascular disease. Consider not using this acronym.

Lines 52-56 has been italicised although reason for this unclear – suggest removing the italic font.

Page 3, paragraph 1 can be split into 2-3 paragraphs. Consider splitting the long paragraphs found throughout the manuscript – it affects the readability of the paper.

Recommend that the aetiology/mechanism sections are accompanied by more graphical representation.

References are often made to the author's work, although not always accompanied by references. It is unclear, in this instance, if the authors are reporting a new unpublished finding.

Comments on the Quality of English Language

Some paragraphs are extremely long and poorly spaced despite different points of interest.

Author Response

Reviewer 1

A very thorough paper although I find that some of the concepts are somewhat repetitive. Overall, the paper is very difficult to read and can be further improved with better paragraphing and keeping the points concise. It appears that the draft is at least 9500-10000 words. I would suggest that a draft half this length should be sufficient to bring forward the key points.

I would like to thank the Reviewer for his comments and recommendations. The Reviewer is undoubtedly right that the paper is extensive, but we wanted to show how many different factors influence the etiology and development of the varicose veins. Chapters that were difficult to read were divided into subchapters to make the text easier to read. The literature was also updated by replacing old citations with new ones.

Other comments:

Figure on page 1 – please provide abbreviations

We added a legend to graphical abstract and an explanation of the abbreviations.

CVD is typically an acronym used for cardiovascular disease. Consider not using this acronym.

We replaced CVD in our work with the term CVI - chronic venous insufficiency

Lines 52-56 has been italicised although reason for this unclear – suggest removing the italic font.

The Reviewer's comment has been taken into account.

Page 3, paragraph 1 can be split into 2-3 paragraphs. Consider splitting the long paragraphs found throughout the manuscript – it affects the readability of the paper.

The Reviewer's proposal was included in the text of the paper.

Recommend that the aetiology/mechanism sections are accompanied by more graphical representation.

In our review, we included three figures related to the etiology/mechanism of varicose vein.

References are often made to the author's work, although not always accompanied by references. It is unclear, in this instance, if the authors are reporting a new unpublished finding.

We included citations from our previous experimental work in the review. There is no new unpublished data in this work.

Reviewer 2 Report

Comments and Suggestions for Authors

The article provides an overview of the works devoted to the pathogenesis of varicose veins. I would like to note that starting from the chapter 5 (‘Oxidative Stress’), this review is an interesting, important, quite complete and understandable presentation of the material. However, the chapters 1 through 4 were difficult to read. There was a violation of the logic of the narrative, a lack of logical connections, and old articles were selected for review. Authors, please pay attention to contemporary articles on the topics you are reviewing.

The comments include requirements and recommendations.

1) The title (lines 2-3), in my point of view, sounds a bit awkward since when the veins become varicose the factors influencing their varicosity do not make sense since they are already varicose (hope it’s not confusing). Probably more appropriate version would be “Factors Influencing Venous Remodeling in the Development of Varicose Veins of the Lower Limbs”.

2) ‘They occur more often in women than in men’ (line 11): Either list all risk factors for varicose veins or remove this one (the gender factor). This looks like the focus of your review, although it is not.

3) Abstract: State the purpose of the work more clearly. The purpose does not correspond to the purpose described in Introduction.

4) ‘Depending on the severity of the disease, varicose veins can be treated with compression stockings, oral medications, injections, laser therapy, scleropathies, or surgery [14]. In some cases, varicose veins can lead to serious consequences such as venous leg ulcers, and thrombosis, including deep vein thrombosis (DVT). Methods used to treat VV, such as venotonic compounds of plant origin and synthetic drugs, are also presented’ (lines 66-70): This sentence is not appropriate here. You are talking about treatment and suddenly interrupted by the consequences.

5) ‘Varicose veins … ultimately ulceration [15]’ (lines 79-92): You should have used a reference to a later review. For example, [https://doi.org/10.3390/jcm10153239].

6) ‘Chronic venous disease is caused by abnormalities in the wall structure and dysfunction of the venous valves, as well as disorders resulting from previous deep vein thrombosis [16]’ (line 92-94); In the lower vessels, an important role for the venous system is performed by the calf muscle pump, referred to as the peripheral heart. With properly functioning valves, contraction of the calf muscle compresses the vein and pumps blood upwards [19]’ (lines 100-103): Refer to a later review.

7) ‘In the lower vessels…’ (line 100): What does this term mean?

8) ‘Changes are also seen in dermal fibroblasts from the same patients, indicating systemic genetic changes’ (lines 109-110): The meaning of this sentence is absolutely not clear. What do you have in mind? What (the same?) patients are you talking about? – the reference [21] is to the article utilizing a model for skin fibrosis. Please re-formulate the sentences (lines 109-112) correctly.  

9) ‘It has been shown that the ratio of collagen I to collagen III is disturbed in CVD patients, i.e., there is an increase in the collagen I/collagen III ratio and MMP-2, and the TIMP-2 expression levels are reduced’ (lines 110-112): There should be a link to the study or review because [21] does not correspond to this conclusion.

10) ‘CEAP1’ (line 138): There is no ‘1’ in the name of this classification.

11) ‘The CEAP1 (Clinical Etiology-Anatomy-Pathophysiology) classification has been adopted throughout the country to enable the assessment of the CVD status and the use of appropriate treatment methods [29]’ (lines 138-140): The CEAP classification has recently been revised [https://doi.org/10.1016/j.jvsv.2019.12.075], so it is advisable to use a later reference.

12) ‘In turn, skin changes and ulcerations are caused by venous hypertension, which is transferred to the microcirculation [30]. However, the cause of skin lesions and ulcers is venous hypertension, which extends to microcirculation’ (lines 146-149): Both of these sentences duplicate each other in meaning.

13) In the chapter 3. Vascular Endothelium (line 156): It will be better if you depict the processes. The sequence of processes is difficult to perceive due to problems with presentation.

14) ‘epithelium’ (line 161): It's better to call it endothelium so as not to be misleading.

15) ‘The epithelium covering the vessels…’ (line 161): Does the epithelium cover blood vessels? Outside? You're talking about the inner layer. Please reformulate the phrase.

16) ‘On the one hand, excessive ROS production inhibits the release of nitric oxide (NO.), which affects vascular stiffness and contractility, leading to endothelial dysfunction [12]. However, the overproduction of NO, leads to oxidative stress and cell apoptosis’ (lines 165-168): Explain which of the two listed processes relates to varicose veins.

17) ‘In addition, endothelial cells regulate gene expression, cell proliferation, and migration’ (line 168-169): What does it mean? How do endothelial cells regulate the listed processes in cells? Which cells exactly do you mean? – the cells within blood (since the endothelium contacts it) or the cells of other venous wall layers? This is written not clearly to understand.

18) ‘Since the oxygenation of the venous wall occurs by diffusion of oxygen from the blood of the lumen and vasa vasorum, the concentration of oxygen in the venous wall is lower in VV than in non-varicose veins’ (lines 249-251): The cause-and-effect relationship is broken here. In this form, the sentence is false, since one does not follow from the other. Please reformulate it.

19) ‘HIF-1α is expressed in most human tissues, possibly all [72]. In contrast, HIF-2α and HIF-3α appear in a small number of tissues (e.g., developing vascular endothelium and fetal lung) and have more limited or specialized roles in oxygen homeostasis [73,74]. In turn, matrix metalloproteinases can lead to inflammation and damage…’ (lines 274-277): You first talked about hypoxia, then suddenly switched to metalloproteinases, as if this were logical; this is misleading and spoils the logic of the presentation.

20) In the chapter 5 (lines 340-343), you are talking about mitochondria in the context of oxidative stress and ROS production, and further, in the chapter 7 (lines 477-483) you are talking about mitochondria in the context of mtDNA released in plasma and bound by RBCs, – all of these INDIRECTLY pointing to varicose veins development; but you don’t even mention the recent work that DIRECTLY connects mtDNA impairments and mitochondria’s dysfunction to varicose veins development [https://doi.org/10.1016/j.vph.2022.107021]. It is advisable to add a brief information and place a reference to this work.

Author Response

Reviewer 2

The article provides an overview of the works devoted to the pathogenesis of varicose veins. I would like to note that starting from the chapter 5 (‘Oxidative Stress’), this review is an interesting, important, quite complete and understandable presentation of the material. However, the chapters 1 through 4 were difficult to read. There was a violation of the logic of the narrative, a lack of logical connections, and old articles were selected for review. Authors, please pay attention to contemporary articles on the topics you are reviewing.

First of all, I would like to thank the Reviewer for carefully reviewing the text of the work as well as for his comments and recommendations. The Reviewer is undoubtedly right that chapters 1 through 4 were difficult to read. These comments have been taken into account. To make reading the text easier, we have divided each chapter into subchapters and updated the literature by replacing old articles with new ones.

All comments made by the Reviewer were also included in the text of the review.

The comments include requirements and recommendations.

  • The title (lines 2-3), in my point of view, sounds a bit awkward since when the veins become varicose the factors influencing their varicosity do not make sense since they are already varicose (hope it’s not confusing). Probably more appropriate version would be “Factors Influencing Venous Remodeling in the Development of Varicose Veins of the Lower Limbs”.

Thank you for your comment related to the title of our work. We introduced the title proposed by the Reviewer.

  • ‘They occur more often in women than in men’(line 11): Either list all risk factors for varicose veins or remove this one (the gender factor). This looks like the focus of your review, although it is not.

This sentence has been removed to avoid implying that it is the main topic of our review.

  • Abstract: State the purpose of the work more clearly. The purpose does not correspond to the purpose described in Introduction.

The summary has been corrected.

  • ‘Depending on the severity of the disease, varicose veins can be treated with compression stockings, oral medications, injections, laser therapy, scleropathies, or surgery [14]. In some cases, varicose veins can lead to serious consequences such as venous leg ulcers, and thrombosis, including deep vein thrombosis (DVT). Methods used to treat VV, such as venotonic compounds of plant origin and synthetic drugs, are also presented’(lines 66-70): This sentence is not appropriate here. You are talking about treatment and suddenly interrupted by the consequences.

Thank you for your comment, which was included in the text of the work.

  • ‘Varicose veins … ultimately ulceration [15]’(lines 79-92): You should have used a reference to a later review. For example, [https://doi.org/10.3390/jcm10153239].

A more up-to-date reference suggested by the Reviewer has been included in the review text.

  • ‘Chronic venous disease is caused by abnormalities in the wall structure and dysfunction of the venous valves, as well as disorders resulting from previous deep vein thrombosis [16]’ (line 92-94); In the lower vessels, an important role for the venous system is performed by the calf muscle pump, referred to as the peripheral heart. With properly functioning valves, contraction of the calf muscle compresses the vein and pumps blood upwards [19]’(lines 100-103): Refer to a later review.

As suggested by the reviewer, the old citation was replaced with a newer one.

  • ‘In the lower vessels…’(line 100): What does this term mean?

Thank you to Reviewer. Of course, it was about the vessels of the lower limbs. Your comment has been included in the review text.

  • ‘Changes are also seen in dermal fibroblasts from the same patients, indicating systemic genetic changes’(lines 109-110): The meaning of this sentence is absolutely not clear. What do you have in mind? What (the same?) patients are you talking about? – the reference [21] is to the article utilizing a model for skin fibrosis. Please re-formulate the sentences (lines 109-112) correctly.

Thank you for your comment. We introduced new sentences and appropriate citations.

  • ‘It has been shown that the ratio of collagen I to collagen III is disturbed in CVD patients, i.e., there is an increase in the collagen I/collagen III ratio and MMP-2, and the TIMP-2 expression levels are reduced’(lines 110-112): There should be a link to the study or review because [21] does not correspond to this conclusion.

We introduced proper citation.

  • ‘CEAP1’(line 138): There is no ‘1’ in the name of this classification.

The Reviewer’s note has been entered in the review text..

  • ‘The CEAP1 (Clinical Etiology-Anatomy-Pathophysiology) classification has been adopted throughout the country to enable the assessment of the CVD status and the use of appropriate treatment methods [29]’(lines 138-140): The CEAP classification has recently been revised [https://doi.org/10.1016/j.jvsv.2019.12.075], so it is advisable to use a later reference.

Thank you for your valuable comment, which we included in the text of the work.

  • ‘In turn, skin changes and ulcerations are caused by venous hypertension, which is transferred to the microcirculation [30]. However, the cause of skin lesions and ulcers is venous hypertension, which extends to microcirculation’(lines 146-149): Both of these sentences duplicate each other in meaning.

Of course, the Reviewer is right, thank you, and one of the sentences has been removed.

  • In the chapter  Vascular Endothelium(line 156): It will be better if you depict the processes. The sequence of processes is difficult to perceive due to problems with presentation.

As suggested by the Reviewer, the sentences have been modified accordingly.

14) ‘epithelium’ (line 161): It's better to call it endothelium so as not to be misleading.

We changed the epithelium to endothelium.

  • ‘The epithelium covering the vessels…’(line 161): Does the epithelium cover blood vessels? Outside? You're talking about the inner layer. Please reformulate the phrase.

As suggested by the Reviewer, the sentences has been changed accordingly.

  • ‘On the one hand, excessive ROS production inhibits the release of nitric oxide (NO.), which affects vascular stiffness and contractility, leading to endothelial dysfunction [12].

As suggested by the Reviewer, the sentence has been changed accordingly.

  • ‘In addition, endothelial cells regulate gene expression, cell proliferation, and migration’(line 168-169): What does it mean? How do endothelial cells regulate the listed processes in cells? Which cells exactly do you mean? – the cells within blood (since the endothelium contacts it) or the cells of other venous wall layers? This is written not clearly to understand.

As suggested by the Reviewer, the sentence has been changed accordingly.

  • ‘Since the oxygenation of the venous wall occurs by diffusion of oxygen from the blood of the lumen and vasa vasorum, the concentration of oxygen in the venous wall is lower in VV than in non-varicose veins’(lines 249-251): The cause-and-effect relationship is broken here. In this form, the sentence is false, since one does not follow from the other. Please reformulate it.

The Reviewer's comment has been taken into account in the text of the review.

  • ‘HIF-1α is expressed in most human tissues, possibly all [72]. In contrast, HIF-2α and HIF-3α appear in a small number of tissues (e.g., developing vascular endothelium and fetal lung) and have more limited or specialized roles in oxygen homeostasis [73,74]. In turn, matrix metalloproteinases can lead to inflammation and damage…’(lines 274-277): You first talked about hypoxia, then suddenly switched to metalloproteinases, as if this were logical; this is misleading and spoils the logic of the presentation.

The Reviewer's comment has been taken into account in the text of the review.

  • In the chapter 5 (lines 340-343), you are talking about mitochondria in the context of oxidative stress and ROS production, and further, in the chapter 7 (lines 477-483) you are talking about mitochondria in the context of mtDNA released in plasma and bound by RBCs, – all of these INDIRECTLY pointing to varicose veins development; but you don’t even mention the recent work that DIRECTLY connects mtDNA impairments and mitochondria’s dysfunction to varicose veins development [https://doi.org/10.1016/j.vph.2022.107021]. It is advisable to add a brief information and place a reference to this work.

We thank the Reviewer for pointing out new research related to mtDNA. We included an excerpt from this publication and a citation in the review text.

Round 2

Reviewer 1 Report

Comments and Suggestions for Authors

I note that the authors are trying to be comprehensive but it is also good to be concise. Assuming no word limitations, I have no further comments

Author Response

We would like to thank the Reviewer for his remark, which we will take into account when preparing the manuscript in the future.

Reviewer 2 Report

Comments and Suggestions for Authors

Dear authors, thank you for replying to all my comments being addressed. The revised version of the manuscript has been considerably improved. Nevertheless, minor correction is needed before the manuscript is accepted for publication.

1) I have one crucial concern about the term CVI that the authors included in the revised version of the manuscript in order to satisfy the recommendation of the 1st reviewer. And namely, that reviewer wrote: “CVD is typically an acronym used for cardiovascular disease. Consider not using this acronym.” The reviewer’s comment in general is correct; it would be hard to disagree with this. The authors replied: “We replaced CVD in our work with the term CVI - chronic venous insufficiency.”

(!) The chosen term (CVI) turned out to be not entirely suitable; I would say it is incorrect in most of the context parts of the manuscript. I’ll try to explain, why. Chronic venous insufficiency (commonly abbreviated as CVI) is a more narrow term (reflecting clinical classes C3-C6, according to the CEAP classification) compared to a more extensive term – chronic venous disease (reflecting all clinical classes C0-C6, according to the CEAP classification). Chronic venous disease includes chronic venous insufficiency but not vice-versa. Apparently, CVI is not always manifested by varicose veins in a patient though edema may be observed. Therefore, after the authors substituted ‘CVD’ with ‘CVI’, many phrases lost their sense. The prominent examples are the sentences in the Abstract: (a) “One of the early symptoms of chronic venous insufficiency (CVI) is varicose veins (VV) of the lower limbs.” (lines 25-26); (b) “The aim of this review was to present the current knowledge on CVI, including the pathophysiology and mechanisms related to vein wall remodeling.” (lines 31-32).

Also, in the lines 155-157 “It has been shown that the ratio of collagen I to collagen III is disturbed in CVI patients, i.e., there is an increase in the collagen I/collagen III ratio and MMP-2, and the TIMP-2 expression levels are reduced [2,21].”  In the cited articles [2,21], the talk is about chronic venous disease but not about chronic venous insufficiency. And so on (I will not continue to list all the cases).

(!!) The appropriate term in the light of the manuscript subject is either (a) chronic venous disease (could be abbreviated as CVeD, and also used in the scientific literature, – that distinguishes it from widely known CVD which stands for cardiovascular disease) or (b) varicose vein disease (could be abbreviated as VVD) or even (c) varicose veins of the lower limbs (could be abbreviated as VVLL), since the authors are mainly talking about varicose veins (the main manifestation of the C2 clinical class, according to the CEAP classification).

I highly recommend reconsidering to use another more appropriate term.

2) In the lines 107-108 there may be missing something (? a word ‘anomalies’ has been used in the previous version): “This review presents the various , in the structure and function of the lower limb veins observed in CVI.”  

3) I have additional comment on the way the authors responded to some of my concerns: they did not specify the line(s) where the changes were introduced. For instance: “Thank you for your comment. We introduced new sentences and appropriate citations.” or “We introduced proper citation.”, or “The Reviewer's comment has been taken into account in the text of the review.” How the reviewer can promptly find the exact places within the modified text where all the lines have been shifted? For the future: after each of such sentences the authors should write an address (line(s)…).

4) In the lines 575-578, it looks like while adding a new reference [137] the authors might misunderstand the meaning (there were no healthy people but VV patients only, and VV vs. NV segments from those patients were investigated), which may mislead the readers. Plus, a decrease in the relative mitochondrial membrane potential in ECs and SMCs were observed in VV.

5) And last, in the line 709, “capers 196” – is this a weight (196g) or a brand name?

Author Response

Reviewer 2

Dear authors, thank you for replying to all my comments being addressed. The revised version of the manuscript has been considerably improved. Nevertheless, minor correction is needed before the manuscript is accepted for publication.

Dear Reviewer, thank you for your comments, recommendations and explanations.

  • I have one crucial concern about the term CVI that the authors included in the revised version of the manuscript in order to satisfy the recommendation of the 1st And namely, that reviewer wrote: “CVD is typically an acronym used for cardiovascular disease. Consider not using this acronym.” The reviewer’s comment in general is correct; it would be hard to disagree with this. The authors replied: “We replaced CVD in our work with the term CVI - chronic venous insufficiency.”

Taking into account the first Reviewer's recommendations, we changed the term CVD to CVI, but reading your review, this was not a good move in all cases.

(!) The chosen term (CVI) turned out to be not entirely suitable; I would say it is incorrect in most of the context parts of the manuscript. I’ll try to explain, why. Chronic venous insufficiency (commonly abbreviated as CVI) is a more narrow term (reflecting clinical classes C3-C6, according to the CEAP classification) compared to a more extensive term – chronic venous disease (reflecting all clinical classes C0-C6, according to the CEAP classification). Chronic venous disease includes chronic venous insufficiency but not vice-versa. Apparently, CVI is not always manifested by varicose veins in a patient though edema may be observed. Therefore, after the authors substituted ‘CVD’ with ‘CVI’, many phrases lost their sense. The prominent examples are the sentences in the Abstract: (a) “One of the early symptoms of chronic venous insufficiency (CVI) is varicose veins (VV) of the lower limbs.” (lines 25-26); (b) “The aim of this review was to present the current knowledge on CVI, including the pathophysiology and mechanisms related to vein wall remodeling.” (lines 31-32).

Thank you for your comments and clarifications on the terminology. These suggestions were also considered in the article by Ortega and colleagues, 2021.

Also, in the lines 155-157 “It has been shown that the ratio of collagen I to collagen III is disturbed in CVI patients, i.e., there is an increase in the collagen I/collagen III ratio and MMP-2, and the TIMP-2 expression levels are reduced [2,21].”  In the cited articles [2,21], the talk is about chronic venous disease but not about chronic venous insufficiency. And so on (I will not continue to list all the cases).

Lines 155-157. Thank you for your attention and comment. We have revised the terminology.

(!!) The appropriate term in the light of the manuscript subject is either (a) chronic venous disease (could be abbreviated as CVeD, and also used in the scientific literature, – that distinguishes it from widely known CVD which stands for cardiovascular disease) or (b) varicose vein disease (could be abbreviated as VVD) or even (c) varicose veins of the lower limbs (could be abbreviated as VVLL), since the authors are mainly talking about varicose veins (the main manifestation of the C2 clinical class, according to the CEAP classification).

I highly recommend reconsidering to use another more appropriate term.

Thank you for your comment. Since the term CVI is chronic venous insufficiency and CVD is a chronic venous disease including cardiovascular disease (hence the first reviewer's suggestion to abandon this term and replace it with the term CVI), we believe that the term CVeD proposed by the Reviewer would be the most appropriate, as it covers quite a broad range of clinical symptoms ranging from varicose veins through swelling, skin lesions to venous ulcers.

  • In the lines 107-108 there may be missing something (? a word ‘anomalies’ has been used in the previous version): “This review presents the various , in the structure and function of the lower limb veins observed in CVI.”  

Thank you for spotting the mistake. The word ‘anomalies’ for some reason had been omitted in the last version we provided. This has also been corrected in the revised version of the manuscript.

  • I have additional comment on the way the authors responded to some of my concerns: they did not specify the line(s) where the changes were introduced. For instance: “Thank you for your comment. We introduced new sentences and appropriate citations.” or “We introduced proper citation.”, or “The Reviewer's comment has been taken into account in the text of the review.” How the reviewer can promptly find the exact places within the modified text where all the lines have been shifted? For the future: after each of such sentences the authors should write an address (line(s)…).

Thank you for your comment. Indeed, we should have provided the line number from the manuscript text with the appropriate answer, comment, etc. We apologize, this was our omission, which resulted from the fact that an inappropriate sentence was deleted in the text of the work and a new, correct one was added. Everything has been marked in yellow, so when reading the text you could easily see the change.

  • In the lines 575-578, it looks like while adding a new reference [137] the authors might misunderstand the meaning (there were no healthy people but VV patients only, and VV vs. NV segments from those patients were investigated), which may mislead the readers. Plus, a decrease in the relative mitochondrial membrane potential in ECs and SMCs were observed in VV.

Thank you for finding this inaccuracy. Indeed, in lines 575-578 described in article [137], these were the same patients whose VV and NV (no-varicose vein) segments were examined. We submitted an amendment.

5) And last, in the line 709, “capers 196” – is this a weight (196g) or a brand name?

In line 709 the word ‘capers’ referred to the plant source containing the greatest amount of rutin. We corrected this sentence by adding "young caper leaves" and including its Latin name. Meanwhile, ‘196’ should have been in the brackets as it was the citation number. Please note that the previous citation has been replaced with a more appropriate one.

Dear Reviewer, we are extremely grateful for your time and all your valuable remarks regarding our updated version of the manuscript. We appreciate your observations which allowed us to make the necessary edits within the text and make the manuscript not only more valuable, but also more understandable and transparent for the future readers. Chapeau bas.